# Event3DGS: Event-Based 3D Gaussian Splatting for High-Speed Robot Egomotion

**Tianyi Xiong**[*], **Jiayi Wu**[*], **Botao He, Cornelia Fermuller, Yiannis Aloimonos,**
**Heng Huang, Christopher A. Metzler**
University of Maryland, College Park
[*] equal contribution
{txiong23, jiayiwu, metzler}@umd.edu

**Abstract:** By combining differentiable rendering with explicit point-based scene representations, 3D Gaussian Splatting (3DGS) has demonstrated breakthrough 3D reconstruction capabilities. However, to date 3DGS has had limited impact on robotics, where high-speed egomotion is pervasive: Egomotion introduces motion blur and leads to artifacts in existing frame-based 3DGS reconstruction methods. To address this challenge, we introduce Event3DGS, an *event-based* 3DGS framework. By exploiting the exceptional temporal resolution of event cameras, Event3GDS can reconstruct high-fidelity 3D structure and appearance under high-speed egomotion. Extensive experiments on multiple synthetic and real-world datasets demonstrate the superiority of Event3DGS compared with existing event-based dense 3D scene reconstruction frameworks; Event3DGS substantially improves reconstruction quality (+3dB) while reducing computational costs by 95%. Our framework also allows one to incorporate a few motion-blurred frame-based measurements into the reconstruction process to further improve appearance fidelity without loss of structural accuracy. The project page is here.

**Keywords:** Event-based 3D Reconstruction, Gaussian Splatting, High-speed Robot Egomotion

## 1 Introduction

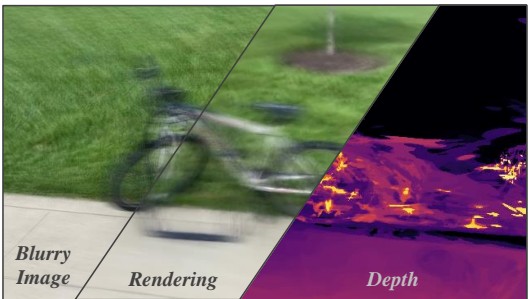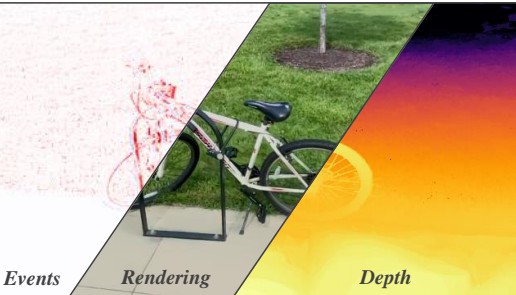

Figure 1: **Left:** Conventional (frame-based) 3D Gaussian Splatting fails to reconstruct geometric details due to motion blur caused by high-speed robot egomotion. **Right**: By exploiting the high temporal resolution of event cameras, Event3DGS can effectively reconstruct structure and appearance in the presence of fast egomotion.

Accurately reconstructing the structure and appearance of 3D scenes from a sequence of 2D images is a fundamental problem in robotics. By combining differentiable rendering models with continuous 3D scene representations, recent "inverse differentiable rendering" (IDR) methods (e.g., Neural Radiance Fields (NeRF) [1] and 3D Gaussian Splatting (3DGS) [2]) have made significant strides in addressing this challenge. Given a sequence of high-quality 2D images, these methods can accurately reconstruct dense 3D geometry and provide near photo-realistic renderings from new views.

However, the accuracy of these methods is fundamentally limited by the quality of their input images: For example, motion blur can severely hamper IDR-based methods ability to reconstruct 3D

8th Conference on Robot Learning (CoRL 2024), Munich, Germany.

geometry. Unfortunately, egomotion-induced motion blur is pervasive in the images captured by real-world robotics systems (e.g., fast-moving drones). Although motion-blur-aware IDR methods have sought to mitigate these effects [3, 4, 5, 6, 7, 8], severe motion blur still fundamentally limits the quality of 3D frame-based reconstructions.

Recent works have sought to overcome these limitations by combining neural radiance fields with event cameras [9, 10, 11, 12, 13, 14]. Event cameras are a novel sensing technology that offers several advantages over frame-based cameras, particularly in the presence of high-speed robot egomotion. By asynchronously recording pixel-level changes in log-intensity, event cameras provide microsecond-level temporal resolution, are robust to motion blur, and have a much higher dynamic range than conventional frame-based cameras [15]. As a result, methods combining neural radiance fields with event data can reconstruct scenes from measurements with substantial egomotion. However, existing methods are impractically slow (hours per reconstruction) and, as we will demonstrate, offer substantial room for improvement with respect to reconstruction accuracy.

In this work we introduce Event3DGS, an event-based 3D reconstruction framework built upon 3D Gaussian Splatting. By integrating the event formation process and differential supervision into the 3DGS framework, Event3DGS recovers multi-view consistent scene representation by minimizing the approximate difference between the integral of observed events and the radiance variations across different rendering views. We also introduce a novel sampling and progressive training strategies to accommodate the sparse characteristics inherent in event data. In addition, Event3DGS can exploit a small number of blurred frame-based images for additional appearance refinement.

Extensive experiments on both simulation and real-world datasets demonstrate that compared to existing event-based IDR methods, Event3DGS can generate comparable or better reconstructions (see Fig. 1) of appearance and geometry while substantially reducing computational costs. Our contributions can be summarized as follows:

1. We introduce a 3D Gaussian Splatting framework for reconstructing appearance and geometry solely from event data.
2. We propose a sparsity-aware sampling and progressive training approaches tailored to event data that improves reconstruction accuracy.
3. We incorporate motion blur into our reconstruction formation process and enable our framework to optionally use motion-blurred frame-based RGB images to improve reconstruction quality.

## 2 Background and Related Work

### 2.1 Novel View Synthesis and 3D Gaussian Splattings

3D scene reconstruction and novel-view synthesis is a fundamental task in graphics and computer vision [16, 17, 18, 19], boosting applications in autonomous driving [20], robotics [21, 22] and virtual reality [23]. NeRF [1] and its variants [24, 25, 26, 27, 28, 29] model a scene implicitly with a MLP-based neural network and utilize differentiable volume rendering, achieving near photo-realistic renderings with high fidelity and fine details. However, since a large number of points need to be sampled to accumulate the color of each pixel, these methods suffer from low rendering efficiency and long training time. Extended works on radiance field aims to accelerate the pipeline by interpolating values from explicit density representations such as points [30], voxel grids [31, 32, 33], or hash grids [34]. Although these methods achieve higher efficiency than the vanilla MLP version, they still need multiple queries for each pixel, lacking real-time rendering capacity.

In light of these challenges, recent research has explored alternative 3D representations for better efficiency and visual fidelity. 3D Gaussian Splatting (3DGS) [2] employs a set of optimized Gaussian splats to achieve state-of-the-art reconstruction quality and rendering speed. Initialized from sparse SfM [35, 36, 37] point clouds, 3DGS is trained via differentiable rendering to adaptively control the density and refine the shape and appearance parameters. A tiled-based rasterizer is proposed to allow for real-time rendering. Multiple works have applied the technique in applications such as SLAM [38, 39], dynamic reconstruction [40, 41], and scene editing [42, 43]. However, all these methods require clear RGB images as input.

## 2.2 Event-based 3D Reconstruction and Radiance Field Rendering

Event-based and event-aided 3D reconstruction [44, 45, 46, 47, 48, 49, 50, 51, 52] and radiance field rendering [53, 9, 54, 55, 56, 57, 13] represent a paradigm shift in computer vision and graphics, enhancing the perception of dynamic scenes with high temporal resolution and accuracy. Weikersdorfer et al. [58] demonstrated event-based stereo reconstruction, illustrating the potential for reconstructing 3D scenes using data from stereo event cameras. However, stereo matching can be challenging due to the sparse nature of event camera data, which often leads to unstable performance in depth estimation [59]. Muglikar et al. [45] enhanced depth sensing by integrating an event camera with a laser projector. While this approach achieves better depth accuracy, the inclusion of a laser projector complicates its effectiveness in outdoor environments with challenging illumination conditions. Previous works introducing event-based radiance fields include Ev-NeRF [53], EventNeRF [9], and E-NeRF [60]. These approaches leverage the inherent multi-view consistency of NeRFs [1], providing a strong self-supervision signal for extracting coherent scene structures from raw event data. However, they inherit NeRF's high computational complexity and challenges in real-time rendering. NeRF's implicit representation complicates editing and integration with traditional 3D graphics processing pipelines.

Our proposed Event3DGS offers explicit, interpretable scene geometry depiction and editable high-fidelity 3D radiance field reconstruction. It allows seamless integration with established graphics pipelines and enables streamlined optimization. Event3DGS is robust under high-speed egomotion, low light, and high dynamic range scenarios where RGB cameras fail to deliver. By combining the event camera's hardware advantages with 3DGS's efficient rendering, our pipeline enables real-time 3D rendering of diverse scenes with low latency, low data bandwidth, and ultra-low power consumption, which supports 3D mapping at a higher operating speed.

## 3 Methodology

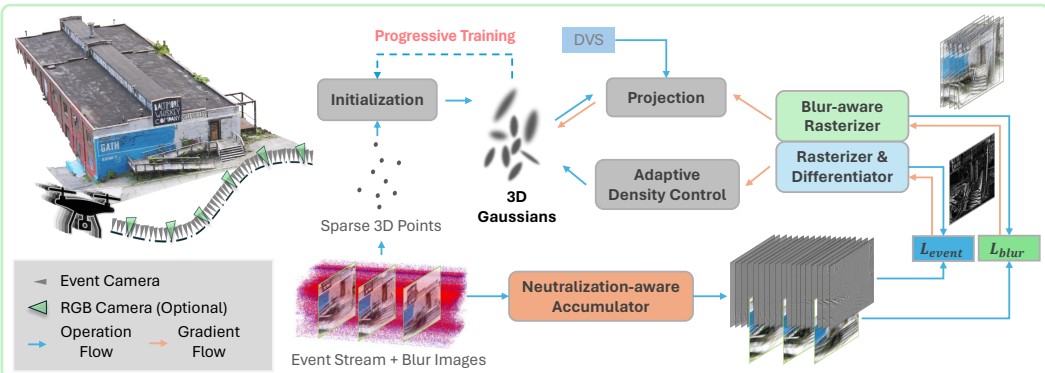

Figure 2: **Event3DGS Architecture**. We first utilize a neutralization-aware accumulator (for mitigating the cancellation of positive and negative events) and sparsity-aware sampling strategy (for reconstruction in non-event regions) to process the input event stream into frames. Then, the sampled event frames are utilized as differential supervision between the corresponding rendered views, optimizing the 3D Gaussians to reconstruct sharp structures and apperance from fast egomotion. We train Event3DGS progressively to better represent geometric details. As an optional component, we integrate a few motion-blurred RGB images from an attached frame-based camera into the pipeline. By embedding motion blur formation into the rasterizer and employing a parameter-separable refinement strategy, we calibrate the colorization while preserving structural details.

The proposed Event3DGS aims to efficiently reconstruct a 3D scene representation from a given sequence of events (either grayscale or color) under high-speed robot egomotion and low-light conditions. Fig. 2 illustrates the overall architecture. Unlike image-based reconstruction, our Event3DGS approach does not directly supervise the absolute radiance of rendered images during optimization. Instead, we integrate the event formation process into the 3DGS pipeline and utilize the observed events as ground truth to implement differential self-supervision within the gradient-based optimization framework. We propose progressive training to further boost reconstruction for fine-

grained structural details. Optionally, to solve the scale ambiguity problem of radiance inherent in event data, we describe parameter separable refinement approach, aligning geometrically sharp Event3DGS with true scene radiance and texture details using a small number of blurred views.

### 3.1 Preliminary

3D Gaussian Splatting (3DGS) [2] explicitly represents a scene with a set of anisotropic 3D Gaussians (ellipsoids). Each Gaussian is defined by a 3D covariance matrix $\Sigma$ with its center point $\mu$:

$$G(x) = e^{-\frac{1}{2}(x-\mu)^T \Sigma^{-1}(x-\mu)} \tag{1}$$

To preserve the valid positive semi-definite property during optimization, the covariance matrix is decomposed into $\Sigma = RSS^T R^T$, where $S \in \mathbb{R}^3_+$ represents scaling factors and $R \in SO(3)$ is the rotation matrix. Each Gaussian is also described with an opacity factor $\sigma \in \mathbb{R}$, and spherical harmonics $\mathcal{C} \in \mathbb{R}^k$ for modeling view-dependent effects.

During optimization, 3D Gaussian Splatting adaptively controls Gaussian density via densification in areas with large view-space positional gradients and pruning points with low opacity. For rendering, the 3D Gaussians $G(x)$ are first projected onto the 2D imaging plane $G'(x)$, then a tile-based rasterizer is applied to enable fast sorting and $\alpha$-blending. The color of pixel $u$ is calculated via blending N ordered overlapping points:

$$C(u) = \sum_{i=1}^{N} c_i \alpha_i \prod_{j=1}^{i-1}(1 - \alpha_j) \tag{2}$$

where $c_i = f(\mathcal{C}_i)$ is the color modeled via spherical harmonics, and $\alpha_i = \sigma_i G'_i(u)$ is the multiplication of opacity and the transformed 2D Gaussian.

### 3.2 Neutralization-aware Slicing & Sparsity-aware Sampling

The input to our Event3DGS pipeline comprises a continuous stream of events $\mathbf{e} = (t, \mathbf{u}, p)$, each indicating a detected increase or decrease in logarithmic brightness (denoted by the polarity $p \in (-1, 1)$) at a specific time instant $t$ and pixel location $\mathbf{u} = (x, y)$. In order to efficiently utilize event data, a common practice is to use event windows to accumulate corresponding events, which requires us to slice the event stream. In event-based 3D radiance field reconstruction pipelines, the slicing strategy of the event stream affects the scene's reconstruction quality. This impact is particularly notable within our pipeline, as neutralization is inevitable during the accumulation of polarity. Existing works [9, 13] have shown that using constant short windows leads to poor propagation of high-level illumination, and using constant long windows often leads to poor local detail. To mitigate the information loss, we design a neutralization-aware event slicing strategy. Our slicing strategy considers the number of events and the neutralization moment to sample the length of the event integration window adaptively by (1) performing slicing when the number of events reaches the threshold, (2) performing slicing where neutralization occurs on many pixels (set threshold manually). This not only ensures the diversity of window lengths but also minimizes the loss of detailed information caused by neutralization.

Uniform radiance regions typically do not trigger events, resulting in spatial sparsity of event data as supervision signals. To mitigate this issue, we introduce low-level Gaussian noise $\mathcal{N}(\mu_{noevt}, \sigma^2_{noevt})$ during the sampling process to augment pixels with no events throughout the entire event window, which enhances the gradient-based optimization on uniform radiance regions and makes our pipeline more robust to noise events. This is expressed formally in Eq. 3:

$$\mathbf{E_u(t_s, t_e)} = \begin{cases} \int_{t_s}^{t_e} \Delta e_{\mathbf{u}}(\tau)d\tau & \text{if \# of event triggers} \neq 0 \\ \Delta \cdot \mathcal{N}(0, \sigma^2_{noevt}) & \text{if \# of event triggers} = 0 \end{cases} \tag{3}$$

where $\mathbf{E_u}$ denotes the accumulation of all event polarities triggered at pixel coordinate $\mathbf{u}$ within the current event window, $\Delta$ is the fixed event threshold, $\sigma_{noevt} = 0.2$ in our experiments, $t_s$ and $t_e$ are the timestamps of the window start and the window end, respectively.

## 3.3 Event Rendering Loss Integrating Structural Dissimilarity

Event data with high temporal resolution provides supervision signals with sharp structural information, allowing 3D Gaussian Splatting (3DGS) to perform fine-grained reconstruction of scene structure under high-speed egomotion. The multi-view consistency of event sequences guarantees the learnable Gaussians to continuously converge to the ground truth geometric structure and logarithmic color field of the scene during optimization. Our event rendering loss $\mathcal{L}_{event}(t_s, t_e)$ compares the recorded events with the differential signal generated by corresponding view renderings according to the event formation model. Following [2], it primarily comprises two components: the $\mathcal{L}_1$ loss, which measures the absolute log-radiance change difference at each pixel, and the structural dissimilarity loss $\mathcal{L}_{DSSIM}$ [61], which accounts for the structural information calculated by neighboring pixels. We define them as follows:

$$\mathcal{L}_1(t_s, t_e) = \left\| \frac{\mathbf{F} \odot (\log \widetilde{\mathbf{C}}(t_e) - \log \widetilde{\mathbf{C}}(t_s))}{g} - \mathbf{F} \odot \mathbf{E}(\mathbf{t_s}, \mathbf{t_e}) \right\|_1 \tag{4}$$

$$\mathcal{L}_{DSSIM}(t_s, t_e) = DSSIM(\frac{\mathbf{F} \odot (\log \widetilde{\mathbf{C}}(t_e) - \log \widetilde{\mathbf{C}}(t_s))}{g}, \mathbf{F} \odot \mathbf{E}(\mathbf{t_s}, \mathbf{t_e})) \tag{5}$$

where $\widetilde{\mathbf{C}}(t)$ denotes the 2D rendering under the view at time $t$, $g$ is a gamma correction value initialized to $2.2$ in our experiments, $\mathbf{E}$ represents the accumulation of all event polarities triggered within the field of view (FOV), $\mathbf{F}$ is the RGGB Bayer filter [9], which only is applied for color events. The total loss can be written as shown in eq. (6), and we set $\lambda_{DSSIM}$ to $0.2$ in our experiments.

$$\mathcal{L}_{event} = (1 - \lambda_{DSSIM})\mathcal{L}_1 + \lambda_{DSSIM}\mathcal{L}_{DSSIM} \tag{6}$$

## 3.4 Progressive Training

The point cloud initialization significantly affects the reconstruction quality of Gaussian Splatting [2, 62]. With precise initial positions, finer structural details can be captured via densification and division of Gaussian splats during training. While Structure-from-Motion (SfM) methods provide accurate point initializations for conventional RGB-based 3D Gaussian Splatting, obtaining precise sparse point initializations directly from event streams is challenging due to the absence of a sufficiently accurate event-based SfM pipeline. Alternatively, we have discovered that Event3DGS, when trained from a random initialization, can itself serve as a relatively accurate initialization. Consequently, we propose a progressive training approach to progressively capture geometric details in under-reconstructed areas. Specifically, given a pretrained Event3DGS that originated from random initialization, we apply an opacity threshold $\alpha_{pro}$ to select Gaussian splats with high opacity and use their center positions as the initialization for the subsequent training rounds. A further detailed illustration is provided in Appendix F.

## 3.5 Blur-aware Rasterization and Parameter Separable Appearance Refinement

Although severely motion-blurred RGB images are challenging for radiance field training due to structural degradation, their true radiance scale and texture information complement event data. We aim to refine the appearance of Event3DGS via training on a small amount of motion blurred inputs, to improve visual fidelity while maintaining sharp scene structure.

In the realm of physics, camera motion blur stems from the amalgamation of radiance induced by the camera's movement. According to the physical image formation, camera motion blur is produced by the integration of radiance during camera movement, which can be mathematically represented with the following equation:

$$\mathbf{I}_{blur} = \int_{\tau_s}^{\tau_e} \mathbf{I}(\mathbf{P}_\tau)\, d\tau \approx \frac{1}{N} \sum_{i=1}^{N} \mathbf{I}(\mathbf{P}_{\tau_i}) \tag{7}$$

where $\mathbf{I}_{blur}$ represents the blurry image, $\mathbf{I}(\mathbf{P}_\tau)$ is the latent sharp image captured at camera pose $\mathbf{P}_\tau \in SE(3)$. To simplify the integral calculation, we approximate it as a finite sum of $N$ radiance values $\mathbf{I}(\mathbf{P}_{\tau_i})$, where $\tau_i$ are the midpoint timestamps of a finite number of event integration windows (EIW) within the exposure interval (from $\tau_s$ to $\tau_e$). To incorporate motion effects due to camera movement during frame capturing into the differentiable rasterization process, we incorporate the above physical formation process of motion blur into the rendering equation:

$$\widetilde{C}_{blur}(x, y, \mathbf{P}_{\frac{\tau_s+\tau_e}{2}}, \mathcal{G}) = \frac{1}{N_{EIW}} \sum_{i=1}^{N_{EIW}} \widetilde{C}(x, y, \mathbf{P}_{\tau_i}, \mathcal{G}) \tag{8}$$

where $\widetilde{C}_{blur}$ denotes the rendered color at pixel (x,y) given by blur-aware rasterization, $\mathcal{G}$ are the 3D Gaussian model parameters, $N_{EIW}$ represents the number of event integration windows within the exposure interval. The loss function $\mathcal{L}_{blur}$ can be written as:

$$\mathcal{L}_{blur} = (1 - \lambda_{DSSIM}) \left\| \widetilde{\mathbf{C}}_{\mathbf{blur}} - \mathbf{I}_{blur} \right\|_1 + \lambda_{DSSIM} DSSIM(\widetilde{\mathbf{C}}_{\mathbf{blur}}, \mathbf{I}_{blur}) \tag{9}$$

To improve the fidelity of scene appearance via a few blurry RGB images while preserving sharp structural details from event sequences, we separate the learnable parameters into two groups. The structure-related parameters include the position $\mu$, scaling factor $S$, and rotation factor $R$; the appearance-related parameters include opacity $\alpha$ and spherical harmonics (SH) coefficients. When trained on event streams, all parameters of Event3DGS are optimized to learn the structure and the approximate logarithmic color field of the target scene. After the parameters have converged on the event stream, we fix the structure-related parameters and only calculate gradients on the appearance-related parameters, using blurry RGB images as supervision to refine the scene's appearance. We scale the learning rate of opacity $\alpha$ by $\eta_\alpha = 0.05$ to inhibit drastic changes in density.

## 4 Experiments

**Synthetic and Real-world Datasets** We evaluate our method using both synthetic and real data. For synthetic scenes, we adopt the dataset proposed in [9], which generates 7 sequences with 360° camera rotations around each 3D object, simulating event streams from 1000 views. For real-world scenes, we first capture videos from a fast-moving RGB camera, then extract frames and estimate camera parameters by COLMAP[35]. We utilize v2e [63] with bayes filter [9] to emulate colorful event streams. The emulated sequences cover both indoor and outdoor scenes under various illumination conditions. We also use the experimental event sequences from [9], which are captured with a DAVIS-346C color event camera on a spinning table illuminated by a 5W light source.

**Metrics and Settings** We report three popular metrics to evaluate our methods: Peak Signal-to-Noise Ratio (PSNR) [64], Structural Similarity Index Measure (SSIM) [61], and AlexNet-based Learned Perceptual Patch Similarity (LPIPS) [65]. Following [9], we apply a linear transformation in the logarithmic space for all our and baseline results. Our implementation is based on the official 3DGS[2] framework. We train our model on a single NVIDIA RTX 6000Ada GPU for $30k$ iterations and filter the Gaussians with opacity threshold $\alpha \geq 0.9$ for progressive training. We randomly initialize the point cloud according to the scale of each training scene and set the other hyperparameters and optimizer as default.

**Baselines** We benchmark our work against a NeRF-based method, EventNeRF [9], and a naive baseline E2VID [66] + NeRF [1], which cascades the event-to-video pipeline E2VID to a vanilla 3D Gaussian Splatting. For synthetic and low-light scenes, we directly render RGB and depth images from the official checkpoints of EventNeRF and only reproduce their training for efficiency evaluation. For real-world scenes, we train EventNeRF for 500k iterations using their default settings. Additional comparisons with deblurring baselines are included in Appendix B.

| Scene | E2VID[66] + 3DGS[2] | | | EventNeRF[9] | | | **Event3DGS (event-only)** | | |
|---|---|---|---|---|---|---|---|---|---|
| | PSNR ↑ | SSIM ↑ | LPIPS ↓ | PSNR ↑ | SSIM ↑ | LPIPS ↓ | PSNR ↑ | SSIM ↑ | LPIPS ↓ |
| Drums | 16.52 | 0.74 | 0.24 | 27.43 | 0.91 | 0.07 | 29.37 | 0.94 | 0.04 |
| Lego | 16.11 | 0.75 | 0.23 | 25.84 | 0.89 | 0.13 | 29.57 | 0.93 | 0.05 |
| Chair | 20.64 | 0.87 | 0.13 | 30.62 | 0.94 | 0.05 | 31.59 | 0.95 | 0.03 |
| Ficus | 23.33 | 0.88 | 0.12 | 31.94 | 0.94 | 0.05 | 32.47 | 0.95 | 0.03 |
| Mic | 20.47 | 0.89 | 0.14 | 31.78 | 0.96 | 0.03 | 33.83 | 0.98 | 0.02 |
| Hotdog | 22.45 | 0.90 | 0.12 | 30.26 | 0.94 | 0.04 | 32.35 | 0.96 | 0.03 |
| Materials | 18.62 | 0.85 | 0.15 | 24.10 | 0.94 | 0.07 | 31.03 | 0.96 | 0.03 |
| Average | 19.73 | 0.84 | 0.16 | 28.85 | 0.93 | 0.06 | **31.46** | **0.95** | **0.03** |

Table 1: Quantitative comparison on synthetic event sequences (event-only). Event3DGS demonstrates best rendering quality across all 7 scenes.

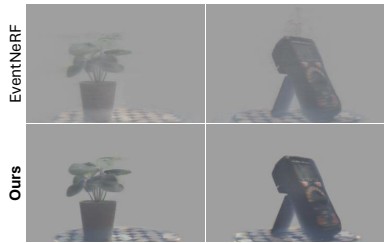

| Scene | EventNeRF[9] | | | Event3DGS (event-only) | | |
|---|---|---|---|---|---|---|
| | PSNR ↑ | SSIM ↑ | LPIPS ↓ | PSNR ↑ | SSIM ↑ | LPIPS ↓ |
| Bike | 21.1 | 0.39 | 0.58 | 23.06 | 0.71 | 0.26 |
| Computer | 20.89 | 0.71 | 0.31 | 24.11 | 0.87 | 0.08 |
| Drum | 21.61 | 0.66 | 0.46 | 24.8 | 0.83 | 0.15 |
| Plant | 16.59 | 0.3 | 0.56 | 22.53 | 0.8 | 0.13 |
| Shoes | 25.35 | 0.78 | 0.39 | 28.08 | 0.89 | 0.16 |
| Average | 21.11 | 0.57 | 0.46 | **24.52** | **0.82** | **0.16** |

Figure 3: Visualization on low-light experimental scenes (event-only).

Table 2: Quantitative comparison on emulated event sequences of real-world scenes (event-only).

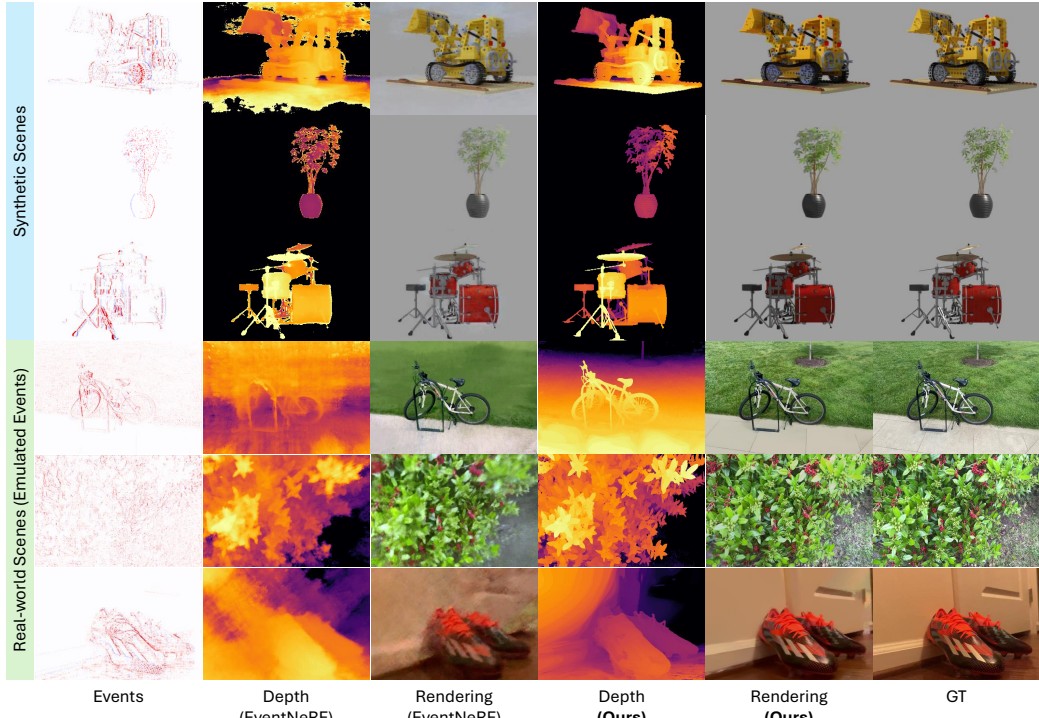

Figure 4: Visualization on synthetic and real-world scenes (events emulated from RGB frames). Event3DGS excels in reconstructing sharp structures and appearance details.

## 4.1 Quantitative Evaluation

**Synthetic Scenes**   As demonstrated in Tab. 1, Event3DGS consistently outperforms both baselines across all synthetic scenes in all metrics. On average, our method achieves a $+2.61dB$ higher PSNR, a $2.15\%$ higher SSIM, and a $50\%$ lower LPIPS.

**Real-world Scenes**   Given that the E2VID [66] + 3DGS [2] baseline performs poorly on forward-looking real-world scenes, we compare our method only with EventNeRF [9]. As shown in Tab. 2, Event3DGS significantly outperforms EventNeRF [9] across all real scenes and metrics, achieving $+3.41$ dB higher PSNR, $43.9\%$ higher SSIM, and $65.2\%$ lower LPIPS on average.

## 4.2 Qualitative Evaluation

We visualize depth maps and renderings on 3 synthetic scenes and 3 real-world scenes. Fig. 4 shows that our method preserves sharper, more consistent structures and cleaner backgrounds compared to EventNeRF [9]. Event3DGS is able to capture detailed information of object edges and geometric discontinuities, such as ficus leaves ($2^{nd}$ row), drum racks ($3^{rd}$ row) and shoelaces ($6^{th}$ row). Our renderings also exhibit higher contrast and sharper details, particularly in highlights and reflections. In the soccer shoe scene, our method captures the reflected lights and corresponding depth, while EventNeRF [9] fails to reconstruct these details. In the bike sample, EventNeRF fails to represent high-frequency details of the grass, whereas our method accurately reconstructs the grass geometry and preserves details in the background. Event3DGS also demonstrates robustness in low-light

conditions. As shown in Fig. 3, our method learns sharper object details (e.g. edges of leaves) with fewer noisy artifacts. We include additional visualization results and deployment on quadrotor in Appendix D and Appendix A respectively.

## 4.3 Ablation Studies and Efficiency Comparison

**Progressive Training** Fig. 5 shows an example of progressive training for improving reconstruction details. With the 3D structure of previous checkpoints, more Gaussians are generated in under-reconstructed areas during the second round of training via adaptive densification. Consequently, Event3DGS is able to progressively capture the subtle details (e.g. bicycle spokes and grasses) that are not accurately modeled during previous rounds.

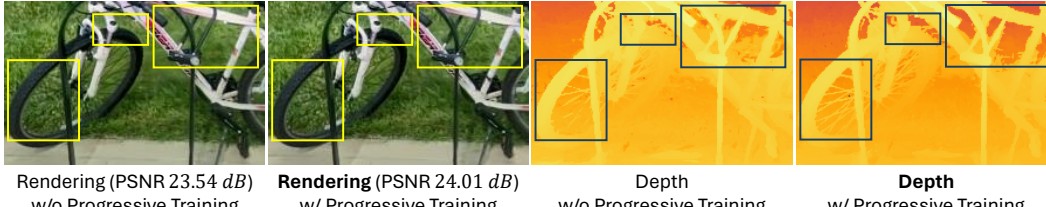

| Rendering (PSNR 23.54 $dB$)
w/o Progressive Training | **Rendering** (PSNR 24.01 $dB$)
w/ Progressive Training | Depth
w/o Progressive Training | **Depth**
w/ Progressive Training |

Figure 5: Ablations on progressive training (event-only). The PSNR we report is for the single image. With the pretrained Gaussians as initialization, Event3DGS is able to progressively recover the fine-grained structural details that are under-reconstructed in the $1^{st}$ round training.

**Blur-aware Appearance Refinement** We adaptively fine-tune appearance-related parameters with $50 - 300$ iterations for each synthetic scene and plot the average PSNR in Fig. 6. As shown, using up to 10 blurry RGB images already yields a noticeable enhancement in rendering quality. We provide more comprehensive ablation studies in Appendix E.

**Model Efficiency.** As shown in Tab. 3, Event3DGS reduces the training time of EventNeRF from 9 hours to less than 20 minutes and achieve over $2000\times$ higher FPS, enabling real-time rendering.

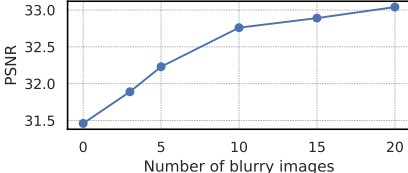

| Method | Synthetic ($346 \times 260$) | | | Real-world ($640 \times 360$) | | |
|---|---|---|---|---|---|---|
| | Training | FPS | Storage | Training | FPS | Storage |
| EventNeRF | 9 hour | 0.5 | 15M | 9 hour | 0.2 | 15M |
| Ours-30k | 6 min | 1018 | 11M | 7 min | 667 | 127M |
| Ours-30k$\times$2 | 12 min | 1036 | 11M | 18 min | 627 | 142M |

Figure 6: Ablations on the number of blurry images.

Table 3: Average model efficiency on synthetic and real-world scenes (event-only).

## 5 Conclusion

Event cameras are a promising tool for sensing and navigating with high-speed robotics. Today, inverse differentiable rendering methods, like EventNeRF [9], are the most effective approach to turn event streams into dense 3D reconstructions. Unfortunately, the computational cost of these methods—hours per scene—make them impractical for most applications. Benefiting from the efficiency of 3D Gaussian Splatting, we present Event3DGS, an event-based 3D dense reconstruction method that achieves state-of-the-art reconstruction quality and significantly accelerate training and rendering. By integrating differential event supervision, sampling, progressive training strategies tailored to event data characteristics, Event3DGS achieves high-fidelity 3D reconstruction under high-speed egomotion and low-light scenarios. Optionally, we introduce parameter-separable fine-tuning to further improve appearance fidelity with a few motion-blurred RGB images, with negligible computational overhead.

This work makes a substantial step towards real-time dense 3D reconstruction with events. By extending the 3D Gaussian Splatting framework to perform reconstruction with event data, our work enables event-based dense 3D reconstructions at a rate $20\times$ faster than existing methods. Still, there remains substantial room for improvement and our method is far from real-time. Further reducing run-times to enable real-time dense 3D reconstruction from events represents an important and exciting direction for future research.

**Acknowledgments**

We thank Shengjie Xu for providing valuable feedback. CAM was supported in part by ARO ECP Award no. W911NF2420113, AFOSR YIP Award no. FA9550-22-1-0208, ONR Award no. N00014-23-1-2752, and NSF CAREER Award no. 2339616. CF acknowledges the support of NSF under grant OISE 2020624. The support of USDA NIFA Sustainable Agriculture System Program under award number 20206801231805 is gratefully acknowledged.

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

## Appendix A   Real-world Quadrotor Experiment

To validate the effectiveness of Event3DGS in real-world robotic applications, we incorporate it into a custom-designed quadrotor platform. As illustrated in Fig. 7(B), we employs an iPhone 13 Pro Max as the data collection device. The drone captures video at 240 FPS with a resolution of 1920 × 1080, which is subsequently converted into an event stream via v2e[63]. We utilize COLMAP[35] to estimate the corresponding camera matrices. Our experimental setting is challenging and aggressive, involving extreme maneuvering conditions: the drone reaches a maximum horizontal acceleration of over 6 $m/s^2$, a maximum roll angular velocity of 87 $deg/s$, and a maximum pitch angular velocity of 48 $deg/s$. Details of these maneuvers are available in our supplementary video.

Experimental results demonstrates that Event3DGS significantly improves both the qualitative and quantitative aspects of event-based 3D dense reconstruction. In Tab. 4, Event3DGS clearly surpasses the baseline across all evaluation metrics. In Fig. 7, Event3DGS accurately reconstructs the sharp geometric structure of the table and trees, whereas EventNeRF[9] cannot preserve those details.

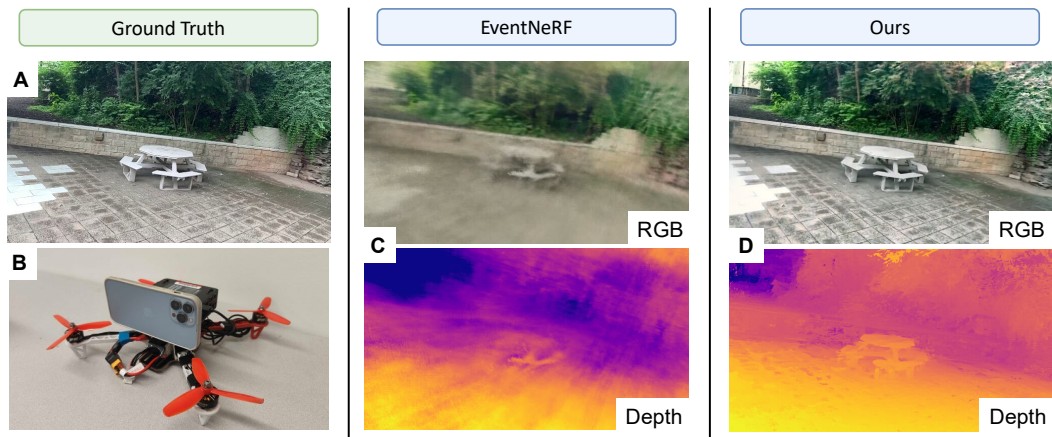

Figure 7: **A:** Ground truth RGB image. **B:** Demonstration of the custom-designed quadrotor. **C:** Rendered RGB and depth of Event3DGS. **D:** Rendered RGB and depth of EventNeRF[9].

| Scene | EventNeRF | | | Event3DGS | | |
|---|---|---|---|---|---|---|
| | PSNR ↑ | SSIM ↑ | LPIPS ↓ | PSNR ↑ | SSIM ↑ | LPIPS ↓ |
| Quadrotor Flight | 16.72 | 0.26 | 0.77 | **19.66** | **0.61** | **0.31** |

Table 4: Quantitative comparison on real-world quadrotor experiment. Due to more complex geometrical structures and larger scale, PSNR of the reported scenes is lower than PSNR of other real-world scenes. However, our method still outperforms EventNeRF[9] by a clear margin.

## Appendix B   Comparision with Deblurring Baselines

In this section, we compare Event3DGS with blur-aware 3DGS baselines: 1) 3DGS + Blur, i.e. vanilla 3D Gaussian Splatting[2] trained with motion-blurred RGB images; 2) DeblurGS[4], a novel method that reconstructs sharp 3D scenes from blurry images via estimating camera motions. We combine the consecutive frames within an event window of length 40 to be a blurry image, and generate 100 blurry training views for each scene. For fair comparison, we set all the hyper-parameters as default for baseline methods.

Since DeblurGS[4] fails to reconstruct the 3D structure of synthetic scenes, we only report the visualization results in Fig. 8. Under high-speed rotations, 3DGS[2] is unable to accurately capture sharp details, and DeblurGS fails to estimate camera motions under severe motion blurs. In contrast, Event3DGS leverages high temporal resolution event data to accurately reconstruct the structure and appearance of the target scene. For real-world scenes, we report the numerical and visualization

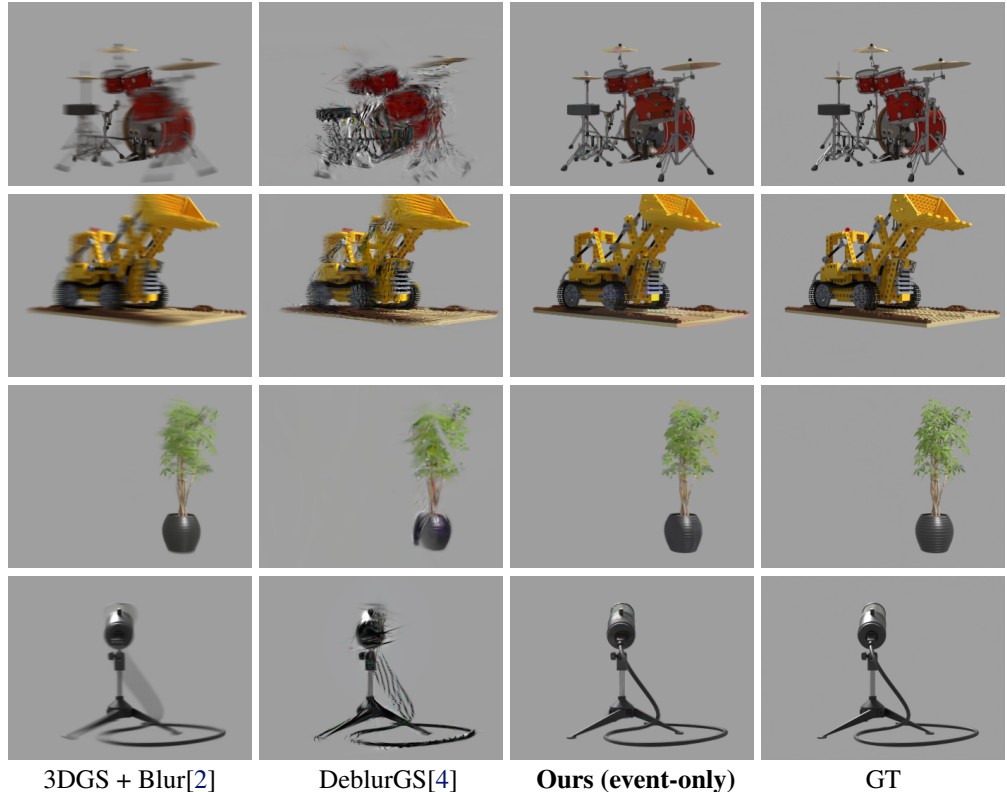

| 3DGS + Blur[2] | DeblurGS[4] | **Ours (event-only)** | GT |

Figure 8: Qualitative comparison with deblurring baselines on synthetic dataset. Data was generated using blender and event simulator [9]. We only report the scenes where rendering of DeblurGS[4] can align with the test views. Event3DGS demonstrates more accurate structural details and better multi-view consistency than baseline methods.

results in Tab. 5 and Fig. 9 respectively. Although DeblurGS roughly deblurs the input images and achieves higher reconstruction quality than the vanilla 3DGS, it fails to preserve multi-view consistency due to the existence of motion blur, causing under-representation in structural details (e.g. bicycle spokes, keyboard, edges of leaves, shoelaces in Fig. 9). As shown in Tab. 5, Event3DGS clearly outperforms baseline methods by an average of $+0.44dB$ higher PSNR, 19% higher SSIM and 33% lower LPIPS.

| Scene | 3DGS[9] + Blur | | | DeblurGS[4] | | | **Event3DGS (event-only)** | | |
|---|---|---|---|---|---|---|---|---|---|
| | PSNR ↑ | SSIM ↑ | LPIPS ↓ | PSNR ↑ | SSIM ↑ | LPIPS ↓ | PSNR ↑ | SSIM ↑ | LPIPS ↓ |
| Bike | 21.0 | 0.42 | 0.62 | 23.90 | 0.54 | 0.42 | 23.06 | 0.71 | 0.26 |
| Computer | 20.75 | 0.64 | 0.42 | 24.58 | 0.80 | 0.13 | 24.11 | 0.87 | 0.08 |
| Drum | 23.79 | 0.68 | 0.41 | 25.48 | 0.76 | 0.18 | 24.8 | 0.83 | 0.15 |
| Plant | 17.05 | 0.34 | 0.57 | 19.28 | 0.52 | 0.28 | 22.53 | 0.8 | 0.13 |
| Shoes | 24.49 | 0.78 | 0.43 | 27.15 | 0.83 | 0.21 | 28.08 | 0.89 | 0.16 |
| Average | 21.42 | 0.57 | 0.49 | 24.08 | 0.69 | 0.24 | **24.52** | **0.82** | **0.16** |

Table 5: Quantitative comparison with deblurring baselines on real-world dataset. Due to the inherent radiance scale ambiguity of event data and the absence of direct color-wise supervision, Event3DGS does not achieve superior PSNR across all scenes. However, it demonstrates the highest structural and perceptual accuracy.

Notably, DeblurGS[8] requires an average of 3.5 hours for training on a synthetic scene due to the high computational cost of motion-blur formation and long training rounds. Event3DGS converges in just 18 minutes with the same hardware (a single NVIDIA RTX 6000Ada GPU), demonstrating significantly higher efficiency.

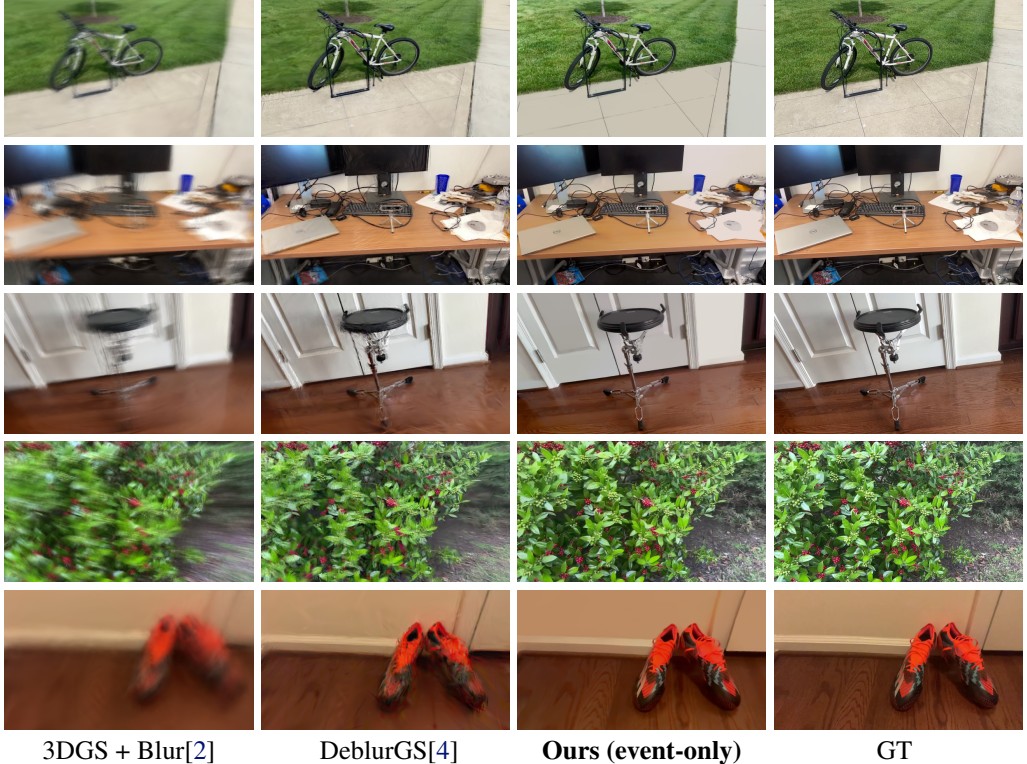

| 3DGS + Blur[2] | DeblurGS[4] | **Ours (event-only)** | GT |

Figure 9: Qualitative comparison with deblurring baselines on real-world dataset. Data was emulated using experimental frame-based data and v2e. Event3DGS reconstructs sharpest details with least motion-blur effects across all scenes.

## Appendix C  Additional Implementation Details

**Real-world Data Capture**   For each real-world scene, we first capture a video from a fast-moving RGB camera, then extract frames and use COLMAP[35] to estimate the corresponding camera extrinsics and intrinsics. We utilize v2e[63] with bayes filter [9] to simulate the colorful event stream.

**Point-cloud Initialization**   Following [2], we start training from 100K uniformly random Gaussians inside a volumetric cube that bounds the scene. For synthetic and low-light sequences proposed in EventNeRF[9], we initialize the scale of points as $l = 0.2$; for our real-world sequences, we set $l = 10$ and move the points to the positive half-axis of $z$.

## Appendix D  Additional Low-light Visualization

For the low-light scenes proposed in [9], objects are placed on a spinning table rotating at a consistent speed of 45 RPM, then event sequences are captured with a DAVIS-346C color event camera under the illumination from a 5W light source. As ground-truth images are not provided in this dataset, we report additional visualization results in Fig. 10. With low-light real sequences, Event3DGS exhibits superior performance in accurately reconstructing sharp geometric details (e.g. edges of the objects) and removing noises on non-event background pixels.

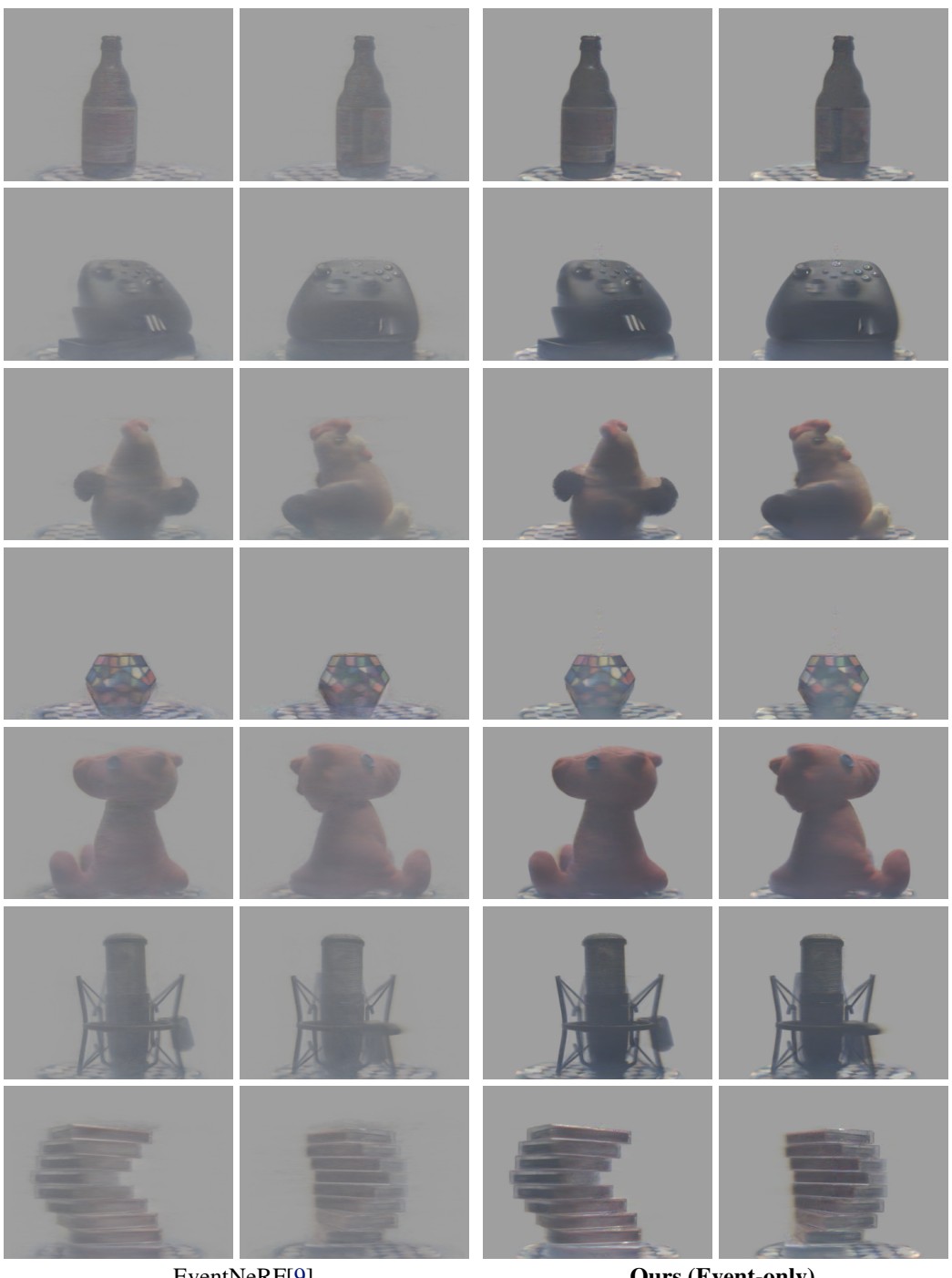

EventNeRF[9]             **Ours (Event-only)**

Figure 10: Visualization results on low-light scenes. Data was experimentally captured using DAVIS-346C[9]. We randomly select two rendered views for each scene. For EventNeRF[9], we directly render images from their official checkpoints.

## Appendix E   Additional Ablation Studies

Here, we present additional ablation studies on each key component of the proposed method to evaluate their individual impacts. All reported results are averaged across all 7 synthetic scenes.

**Ablation on Loss Functions**   As demonstrated in Tab. 6 and Fig. 11, using only the L1 loss results in a lack of detailed textures, while relying solely on the DSSIM loss leads to inaccurate color variations and artifacts. Utilizing both L1 and DSSIM losses together achieves the best performance in reconstructing both appearance and structural details.

| Loss Function | PSNR ↑ | SSIM ↑ | LPIPS ↓ |
|---|---|---|---|
| L1 + DSSIM (ours) | **31.46** | **0.95** | **0.03** |
| L1 only | 29.22 | 0.94 | 0.08 |
| DSSIM only | 29.28 | 0.94 | 0.04 |

Table 6: Ablation on loss functions.

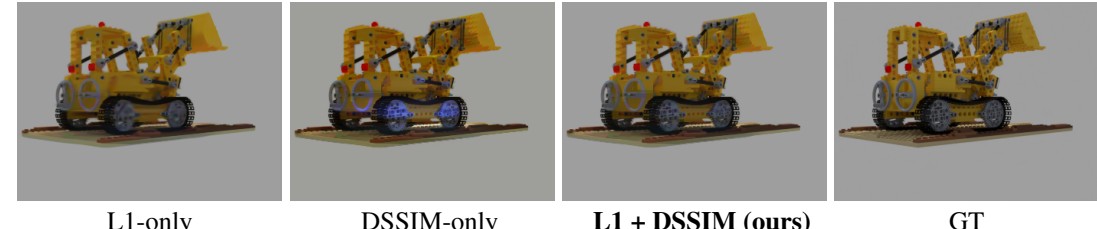

| L1-only | DSSIM-only | **L1 + DSSIM (ours)** | GT |

Figure 11: Visualization results based on choices of loss functions

**Ablation on Slicing Strategy**   For all experiments, we do not apply progressive training and only conduct $1^{st}$ round training for 30k iterations. EventNeRF[9] applies randomized length slicing and negative sampling. As shown in Tab. 7, our slicing strategy leads to overall performance gain, whereas simply applying EventNeRF's strategy onto Gaussian Splatting does not result in satisfactory improvement.

| Slicing Strategy | PSNR ↑ | SSIM ↑ | LPIPS ↓ |
|---|---|---|---|
| Ours (w/o progressive training) | **30.92** | **0.95** | **0.04** |
| EventNeRF[9] | 30.49 | 0.94 | **0.04** |
| Fixed $window\_length = 30$ | 30.40 | 0.94 | 0.05 |

Table 7: Ablations on slicing strategies.

**Ablations on $\sigma_{noevt}$**   This parameter represents the scale of gaussian noise we add to the non-event pixels. As shown in Tab. 8, while it is not highly sensitive, $\sigma_{noevt} = 0.2$ works best in our experiments.

| $\sigma_{noevt}$ | PSNR ↑ | SSIM ↑ | LPIPS ↓ |
|---|---|---|---|
| 0 | 29.69 | 0.94 | 0.06 |
| 0.1 | 31.05 | **0.95** | 0.04 |
| 0.2 (ours) | **31.46** | **0.95** | **0.03** |
| 0.5 | 31.34 | **0.95** | **0.03** |
| 1.0 | 29.84 | **0.95** | 0.05 |

Table 8: Ablations on $\sigma_{noevt}$.

**Ablations on Progressive Training** As shown in Tab. 9, progressive training leads to further improvement in PSNR, whereas merely increasing the number of training iterations does not yield better results. While increasing the number of rounds lead to marginal performance gain, we report the results of 2-round progressive training as our final results, to balance between performance and time efficiency.

| Training Iterations | PSNR ↑ | SSIM ↑ | LPIPS ↓ |
|---|---|---|---|
| 30k | 30.92 | **0.95** | 0.04 |
| 60k | 30.99 | **0.95** | 0.04 |
| 30k * 2 rounds(ours) | **31.46** | **0.95** | **0.03** |
| 30k * 3 rounds(ours) | **31.50** | **0.95** | **0.03** |

Table 9: Ablations on progressive training.

## Appendix F    Detailed Explanation of Progressive Training

We illustrate the process of 2-round progressive training using the following pseudo code:

1. **Initialize Event3DGS with randomized points:**

$$G_{10} \leftarrow G^{(0)} = (X^{(0)}, \square^{(0)})$$

where $X^{(0)} = \{\mu_i^{(0)} | \mu_i^{(0)} \overset{\text{iid}}{\sim} \mathbb{R}^3\}_{i=1}^{N_0}$ represents the center positions of the Gaussians, $\square^{(0)} = \{(S_i^{(0)}, R_i^{(0)}, \mathcal{C}_i^{(0)}, \sigma_i^{(0)})\}_{i=1}^{N_0}$ includes other parameters (scaling factor $S_i$, rotation factor $R_i$, spherical harmonics $\mathcal{C}_i$, and opacity $\sigma_i$), all of which are randomly initialized.

2. **For the $1^{st}$ round, train the Event3DGS to minimize the event rendering loss:**

$$G_1^* = (X_1^*, \square_1^*) = \arg\min_G \mathcal{L}_{event}(G_1)$$

3. **Select the gaussians with high opacity and apply their positions to be the initialization for the $2^{nd}$ round:**

$$G_{20} \leftarrow G^{(1)} = (X^{(1)}, \square^{(1)})$$

where $X^{(1)} = \{x_{1j} \mid x_{1j} \in X_1^* \wedge \sigma_{1j} > \alpha_{pro}\}$ is the set of points with high opacity from the first-round checkpoint, and $\square^{(1)}$ is randomly initialized.

4. **Progressively train the Event3DGS for the $2^{nd}$ round:**

$$G_2^* = (X_2^*, \square_2^*) = \arg\min_G \mathcal{L}_{event}(G_2)$$

Repeating step 3 and step 4 results in multiple rounds of progressive training.

