# OpenReview forum: "Event3DGS: Event-Based 3D Gaussian Splatting for High-Speed Robot Egomotion"
_robot-learning.org/CoRL/2024/Conference — CoRL 2024_

### Official Review · Reviewer_a756 · 2024-07-19
**Good result but marginal contribution on top of the formulation of EventNerf + 3DGS backbone and no explanation on why it works better.**

**Originality:** 2
**Technical Quality:** 3
**Clarity Of Presentation:** 4
**Potential Impact:** 2
**Recommendation:** 3
**Confidence:** 3

**Review:**

### Strengths

- Good performance in various datasets.
- Significantly faster in training/inference thanks to 3DGS.
- Easy to read and straight forward method.

### Weaknesses

- Not much technical contribution and explanation.
  - The formulation is very similar to the baseline method (EventNerf [9]). Why is the propose method works better? Could you provide equivalent version of Nerf or 3DGS trained with the EventNerf loss?
  - If the different loss is the contributing factor, the ablation study should be included.
  - Comparison to different slicing approaches to verify the proposed approach is contributing to the performance (e.g. EventNerf uses randomized length slices).
- Important sections do not have detailed explanation.
    - How do you get the camera poses and are they updated during training? Are they accurate from blurry images? If the poses are relatively accurate, can we use the sparse pointcloud from the pose estimation as the initial points?
    - Progressive training: how is this different from the conventional 3DGS training?
    - How are the parameters are chosen: $\sigma_{noevt}$, gamma correction parameter $g$, and $\Delta$ and how sensitive they are?
- Needs more baselines or experiments.

**Quality Of The Limitations Section:**

2

**Questions For Rebuttal:**

Most of the questions are addressed in the review section. One additional question: how is $g$ global parameter? Are all the data use the same event camera? If $g$ is a coefficient to convert from integrated events to an image, my understanding is that this should be varying by camera.

**Robotics Focus:**

4

**Summary Of Paper:**

The paper proposes an event-based 3DGS that leverages the high-temporal resolution of the event camera.

**Summary Of Recommendation:**

Even with lack of explanations, it combined EventNerf and 3DGS with proper adaptations and proposed a faster and better model.

---

### Official Review · Reviewer_AkQ7 · 2024-07-19
**Replacing EventNeRF’s NeRF with Gaussian Splatting**

**Originality:** 2
**Technical Quality:** 2
**Clarity Of Presentation:** 3
**Potential Impact:** 1
**Recommendation:** 3
**Confidence:** 4

**Review:**

Positives:
+ The empirical results are certainly promising. The proposed approach is both, faster and more accurate than the relevant baseline.
+ Results are presented across both, real-world and synthetic settings.
+ The presented approach allows for a minor modification to handle motion blur, and this is show to be effective (although this is orthogonal to the gaussian splatting)
+ The ablations do show the benefit of progressive initialization

Concerns:
- The technical contributions are a bit limited. In essence, the paper seeks to replace the 3D representation in a prior work to a currently more popular one, and there really aren't any interesting technical innovations in doing so.

- The connection to robotics is rather tenuous. The experiments are all in synthetic or self-collected setups, and not really in a scenario that a reader can directly connect to robotics. Moreover, the approach assumes known camera poses, which itself is a huge challenge in high-speed ego-motion settings.

As a minor point, the paper title should replace ‘for’ with ‘under’ because the paper does not really show any use of the reconstruction for the high-speed motion.

**Quality Of The Limitations Section:**

2

**Questions For Rebuttal:**

- The GS paper had two losses in RGB space which this work adopts, but are both these still needed even with EventGS?
- How are the test viewpoints selected? Are these a subset of the training viewpoints (albeit with the caveat that their RGB images were not directly used)?

**Robotics Focus:**

1

**Summary Of Paper:**

This paper tackles the task of 3D reconstruction using event-based cameras. Following EventNeRF, which used a loss between the rendered and integrated differential signals, this work trains a 3D scene representation. The key contribution is to replace the NeRF backend with a 3D Gaussian Splatting one while figuring out key implementation details e.g. progressive training instead of relying on SfM initialization. The resulting approach is significantly faster than EventNeRF while also being more accurate.

**Summary Of Recommendation:**

Overall, while the empirical results for reconstruction/view synthesis are promising, the lack of technical contributions and questionable relevance to robotics prevent a recommendation towards accept. Edit: Leaning towards accept after author response.

---

### Official Review · Reviewer_VHGX · 2024-07-21
**Significant step in performance, writing is lacking.**

**Originality:** 3
**Technical Quality:** 3
**Clarity Of Presentation:** 2
**Potential Impact:** 3
**Recommendation:** 3
**Confidence:** 4

**Review:**

This paper addresses an interesting problem: 3D reconstruction using DVS cameras, and optionally RGB frames. This is a meaningful problem in robotics as fast ego-motion can make 3D reconstruction and SfM pipelines fail. Another application not mentioned by the authors is high-speed, dynamic manipulation, which is particularly relevant in industrial settings. Event-based cameras are one possible solution.

The results are good and show Event3DGS outperforms the SoA approaches, both quantitatively and qualitatively. I do have some concerns with this paper:
- The method is not explained clearly. For example, section 3.4 on progressive training only states a pre-trained Event3DGS is used. What does this mean? This is similar in other places in the paper.
- No method ablations are presented. It would be great to see the impact of the contributions, and this would show if a more  simple version would achieve the same result.

There are also some papers not mentioned which I believe should be part of the camera ready:
"EvaGaussians: Event Stream Assisted Gaussian Splatting from Blurry Images" - https://arxiv.org/abs/2405.20224
"Ev-GS: Event-based Gaussian splatting for Efficient and Accurate Radiance Field Rendering" - https://arxiv.org/abs/2407.11343 - this work came out after the submission deadline and does not have to be considered a baseline.

**Quality Of The Limitations Section:**

1

**Questions For Rebuttal:**

Please consider my comments above.

**Robotics Focus:**

3

**Summary Of Paper:**

This paper proposes a new method for leveraging event-based cameras, or DVS (dynamic vision sensor) cameras, to enable 3D reconstruction. While prior works used NeRF representations for a similar task, this work proposes to leverage Gaussian splats. The contributions are: neutralization-aware slicing, sparsity-aware sampling, blur-aware rasterization and separable apperance refinement. The results show a significant improvement in quality compared to related baselines.

**Summary Of Recommendation:**

To the best of my knowledge, this is currently the SoA for 3D reconstruction using event-based cameras. While the writing might be lacking and the method is not entirely novel, I believe it should be accepted. It has clear applications in robotics.

---

### Author Rebuttal · Authors · 2024-08-11

We would like to thank the reviewers and area chair for their efforts and time in providing thoughtful feedback and comments, which significantly helped to improve the quality of our work. We are greatly encouraged by the reviewers' recognition and positive evaluation of our work, including:

- 3D dense reconstruction with event camera is crucial in robotics, as rapid ego-motion can cause frame-based 3D reconstruction and SfM pipelines to fail. The proposed event-based method can be a viable solution for high-speed, dynamic manipulation which is relevant in industrial applications.(VHGX)
- Results on both real-world and synthetic data are promising.(VHGX, AkQ7) The proposed approach is both, faster and more accurate than the relevant baselines.(AkQ7, a756)
- Progressive training is effective and shows clear benefits in the ablation studies.(AkQ7)

We thank the reviewers for pointing out the shortcomings in our writing and the need for additional ablation studies. In response to the feedback, we provide individual responses to each reviewer's concerns below and revise the manuscript, highlighting changes in blue for clarity and additional discussion. We include all additional ablation studies and analyses in the Appendix. Briefly, we summarize the additional experiments and revisions made to the paper:

- Revision of Section 3.4 on progressive training, and a detailed illustration in Appendix F.
- Additional ablation studies in Appendix E to show the effectiveness of each individual component, including:
    -  loss functions
    -  slicing strategies
    -  choices of parameters and how sensitive they are
    -  progresive training

For more details, please refer to the individual responses. We hope our responses will effectively addressed your concerns. Please don’t hesitate to let us know of any additional comments or feedback on improvement.

---

### Decision · Program_Chairs · 2024-09-04

**Decision:**

Accept

**Comment:**

Summary: This paper proposes Event3DGS which  combines differentiable rendering with event cameras to enhance 3D reconstruction under high-speed egomotion, significantly improving quality and efficiency compared to existing methods.

Strengths:
* Shows superior performance over state-of-the-art approaches, both quantitatively and qualitatively.
* Significantly faster training and inference due to the use of 3D Gaussian Splatting (3DGS).
* Incorporates a method to handle motion blur, enhancing appearance fidelity without compromising structural accuracy.

Weakness:
* Key sections, such as progressive training, lack detailed explanation.
* Insufficient comparison to baseline methods like EventNerf; needs more detailed analysis to explain why the proposed method works better.
* No ablation study on the impact of individual components.
* Experiments do not directly connect to real-world robotics scenarios, and the assumption of known camera poses is a limitation in high-speed ego-motion settings.
* This paper fits more for a computer vision conference paper than CoRL.

----

Post rebuttal: The reviewers are happy with the authors' replies and have reached a consensus on accepting the paper.